# Challenges with pediatric antiretroviral therapy administration: Qualitative perspectives from caregivers and HIV providers in Kenya

Michala Sliefert[1], May Maloba[2], Catherine Wexler[1]*, Frederick Were[3], Yvonne Mbithi[3], George Mugendi[3], Edward Maliski[4], Zachary Nicolay[4], Gregory Thomas[5], Shadrack Kale[2], Nicodemus Maosa[2], Sarah Finocchario-Kessler[1]

1 University of Kansas Medical Center, Kansas City, KS, United States of America, 2 Global Health Innovations, Nairobi, Kenya, 3 University of Nairobi, Nairobi, Kenya, 4 Oak Therapeutics, Lawrence, KS, United States of America, 5 University of Kansas, Lawrence, KS, United States of America

* Cwexler@kumc.edu

**Data Availability Statement:** Data cannot be shared publicly, per restrictions set in place by the Institutional Review Boards at the University of

## Abstract

### Background

Current formulations of pediatric antiretroviral therapy (ART) for children with HIV present significant barriers to adherence, leading to drug resistance, ART ineffectiveness, and preventable child morbidity and mortality. Understanding these challenges and how they contribute to suboptimal adherence is an important step in improving outcomes. This qualitative study describes how regimen-related challenges create barriers to adherence and impact families.

### Methods

We conducted key informant interviews (KIIs) with 30 healthcare providers and 9 focus group discussions (FGDs) with a total of 72 caregivers, across three public hospitals in Siaya and Mombasa Kenya. The KIIs and FGDs were audio recorded, translated, and transcribed verbatim. The transcripts were hand coded based on emergent and a-priori themes.

### Results

Caregivers discussed major regimen-related challenges to adherence included poor palatability of current formulations, complex preparation, and administration (including measuring, crushing, dissolving, mixing), complex drug storage, and frequent refill appointments and how these regimen-related challenges contributed to individual and intrapersonal barriers to adherence. Caregivers discussed how poor taste led to child anxiety, refusal of medications, and the need for caregivers to use bribes or threats during administration. Complex preparation led to concerns and challenges about maintaining privacy and confidentiality, especially during times of travel. Providers corroborated this patient experience and described how these challenges with administration led to poor infant outcomes, including

Kansas Medical Center and University of Nairobi. Data can be requested at mdiniz@kumc.edu.

**Funding:** This study was funded by the National Institute of Child Health and Human Development, award number R21HD105534 awarded to Dr. Sarah Finocchario-Kessler. The funders had no role in study design, data collection and analysis, decision to publish, or preparation of the manuscript.

**Competing interests:** The authors have declared that no competing interests exist.

high viral load and preventable morbidity. Providers discussed how the frequency of refills could range from every 2 weeks to every 3 months, depending on the patient. Caregivers discussed how these refill frequencies interrupted work and school schedules, risked unwanted disclosure to peers, required use of financial resources for travel, and ultimately were a challenge to adherence.

## Conclusion

These findings highlight the need for improved formulations for pediatric ART to ease the daily burden on caregivers and children to increase adherence, improve child health, and overall quality of life of families.

## 1.0 Introduction

The World Health Organization (WHO) guidelines call for all infants and children living with HIV to initiate antiretroviral therapy (ART) immediately after diagnosis [1]. Globally, 1.7 million pediatric patients (<15 years) require such treatments [2]. In Kenya, current first line ART regimens include AZT+3TC+NVP for children under 4 weeks of age (available in syrup), ABC+3TC+LPV/r for children 4 weeks to 3 years of age (ABC/3TC is available in tablet only, while LPV/r is available in syrup or table), and ABC+3TC+EFV for children 3–14 years of age (available in tablets, only) [3]. These current pediatric ART formulations pose significant barriers to ART adherence which contribute to ART ineffectiveness, drug resistance, and preventable child morbidity and mortality [4–6].

Regimen-related barriers to pediatric ART adherence affecting caregivers include complex dosing and administration requirements and risk of stigma and unintentional disclosure of HIV status. Current formulations require caregivers to measure and then prepare several drugs per dose which creates multiple opportunities for dosing errors or spillage that may hinder treatment efficacy [7]. This process is both time-consuming and conspicuous [7, 8]. This can result in inadvertent HIV status disclosure and stigma if friends or family discover the routine. Frequent weight and age related dosing changes further increase complexity [1] Primary regimen-related barriers affecting child/ infant ART adherence are inability to safely ingest solid tablets and the poor palatability of liquid formulations as either syrups or tablets reconstituted with water or breastmilk [7–11]. It is difficult to ensure delivery of an effective dose because infants and young children often resist ingestion, and can spit out liquid formulations administered via syringe [10].

Pediatric ART syrups are packaged in large bottles that are conspicuous when leaving the hospital, traveling, or storing within the home [7]. In many lower resource areas, including Kenya, ABC+3TC syrups (fixed-dose combination antiretroviral medication of Abacavir and Lamivudine, current first line pediatric medication for children <30kg in Kenya) [12] are not reliably available, thus tablets are crushed and dissolved in water as part of the treatment regimen. In some cases, this also requires boiling water to make it potable, adding to the burden and cost (for fuel) of ART preparation and administration for children. In settings with high HIV stigma and low disclosure rates, the current standard of pediatric ART care creates many barriers that often result in missed doses and low adherence, which can result in poor clinical outcomes for the child [7].

Our team is exploring alternative delivery strategies for pediatric ART and sought to understand challenges experienced with the regimens currently available in Kenya. Here, we

describe how regimen-related challenges to pediatric ART pose challenges to families. This study is part one of a three-phase study with the overall objective of informing the design and development of alternative delivery mechanisms for pediatric ART regimens that will mitigate current barriers to improve adherence and optimize child health and survival. Next phases of the study will include integrating feedback into formulation and packaging prototypes and seeking stakeholder feedback on prototypes to finalize design elements.

## 2.0 Methods

This study describes challenges of ART administration for caregivers and providers through caregiver focus groups and provider key informant interviews (KIIs). Caregivers were eligible to participate in focus group discussions if they cared for a child living with HIV less than 10 years old and received HIV care for their child at one of the study hospitals (Bondo Referral Hospital in Siaya County, Ambira Sub-County Hospital in Siaya County, and Port Reitz Sub-County Hospital in Mombasa County). Both facility leaders and county ministries of health were sensitized on study goals and procedures and approved the study prior to implementation. These facilities were chosen because our team had established study infrastructure and working relationships there and they represented various health facility levels and regions. Caregivers were recruited by mentor mothers (mothers living with HIV who have been through PMTCT and EID services and serve as peer health advisors) or providers with established clinical care relationships with clients at the study sites. Providers were eligible for interviews if they routinely participated in pediatric HIV care (including counseling, ART prescription or dispensing, HIV clinical care) at one of the study sites. Participants were purposefully selected to represent caregivers over a range of child ages and providers with a variety of HIV care provision roles within the hospital. No eligible caregiver or provider refused participation in the study.

Participants provided written informed consent in compliance with IRB approvals at the University of Kansas Medical Center (STUDY00147024) and University of Nairobi (P457/06/2021). Caregiver informed consent documents and FGD guides were available in English, Luo, and Swahili and all documents and translations were approved by the IRB at the University of Nairobi. Provider informed consent documents and interview guides were available in English only, as all healthcare providers were fluent in English. During the informed consent process, the interviewer explained the purpose of the study, risks and benefits of the study, and emphasized that study participant was optional. Immediately after informed consent was provided but prior to the focus groups or KIIs, surveys with each participant were conducted to capture socio-demographics characteristics of providers and caregivers. Informed consent interviews and surveys occurred in a private area of the hospital and were conducted by a research assistant in English, Swahili, or Luo, per the participant's preference. Focus groups and KIIs lasted approximately 90 or 30 minutes, respectively. Participants were provided $10 remuneration in appreciation for their time and contributions and soda was to each provided to each participant during the KII and FGD

Focus groups and interviews occurred from February to April of 2022. Focus groups and interviews were conducted by authors and study managers SB, NM, or a research assistant in a private area of the study hospital. Only the participant(s) and facilitator were present during the FGD or KII. Providers had interacted with study coordinators SB and NM prior to the interviews as part of other, ongoing research studies; however, caregivers did not have a previous relationship with the interviewers. All interviewers received training on the purpose of the study and qualitative methods. No other authors had access to participant identifying information.

Focus groups and interview guides probed about challenges with the current pediatric anti-retroviral regimen. Guides were reviewed by Kenyan pediatricians (FW, YM) and nurses (MM) for acceptability. Focus groups and interviews were audio recorded and transcribed. For focus groups conducted in Kiswahili or Luo, transcripts were translated into English. Two analysts (MS and CW) coded transcripts based on a priori themes related to how characteristics of the ART regimen created individual-level and interpersonal barriers to adherence. Two analysts met periodically to refine the codebook and develop consensus. Analysts coded the transcripts and developed memos for each code to summarize the most salient themes, in this case challenges with HIV medication regimen. Excel was used to organize codes. Field notes were not taken, and transcripts were not returned to participants for comment or correction.

## 3.0 Results

A total of 9 focus groups were conducted with 72 total caregivers (mean age 37.3 years old, range 18–74) who provided care for n = 83 children (mean age 4.7 years old, range 5 months-18 years old; the caregiver to the 18-year-old also cared for younger children) living with HIV across three hospitals in Kenya. The majority of caregivers were parents (60.6%) and the majority of caregivers were female (83.3%). Twenty out of the 83 children cared for (24%) knew their HIV status. About 76% of the caregivers were living with HIV. Additionally, 30 KIIs were conducted with healthcare providers who provide pediatric HIV care across the same three hospitals (10 providers per hospital). The healthcare providers included clinical officers, mentor mothers, community health volunteers, nurses, and others, with a mean time in healthcare of 9.85 years. See Tables 1 and 2 for full caregiver and provider characteristics. Table 3 outlines key characteristics of caregiver FGD.

**Table 1. Caregiver characteristics.**

| Gender | N | % |
|---|---|---|
| males | 12 | 16.7% |
| Females | 60 | 83.3% |
| Relationship to Child | | |
| Grandparent | 14 | 19.7% |
| Aunt | 8 | 11.3% |
| Parent | 43 | 60.6% |
| Other | 1 | 1.4% |
| Sibling | 3 | 4.2% |
| Step mother | 2 | 2.8% |
| Non-family treatment support | 1 | 1.4% |
| Number of HIV+ Children in Care | | |
| 1 | 61 | 84.7% |
| 2 | 10 | 13.9% |
| 3 | 1 | 1.4% |
| Child's Disclosure Status | | |
| Disclosed status to anyone but caregiver | 69 | 95.8% |
| "I am the only one who gives the child his/her ART" | | |
| TRUE | 23 | 31.9% |
| Weekly household income | | |
| <500 | 32 | 44.4% |
| 500–750 | 18 | 25.0% |
| 750–1000 | 9 | 12.5% |

*(Continued)*

**Table 1.** (Continued)

| | | |
|---|---|---|
| 1000–2500 | 9 | 12.5% |
| >2500 | 4 | 5.6% |
| **Level of education** | | |
| No school | 2 | 2.8% |
| Some Primary | 24 | 33.3% |
| Completed Primary | 19 | 26.4% |
| Some Secondary | 11 | 15.3% |
| Completed Secondary | 13 | 18.1% |
| Some college/university | 3 | 4.2% |
| **Caregiver HIV Status** | | |
| HIV+ | 55 | 76.4% |
| HIV- | 16 | 22.2% |
| Unknown | 1 | 1.4% |

**Table 2. Provider characteristics.**

| Sex | | |
|---|---|---|
| Female | 19 | 63.3% |
| Provider role | | |
| Mentor Mother | 3 | 10.0% |
| Pharmaceutical Technologist | 3 | 10.0% |
| CHV | 3 | 10.0% |
| Nurse | 7 | 23.3% |
| Clinical Officer | 6 | 20.0% |
| Hospital Administrator | 3 | 10.0% |
| County Administrator | 1 | 3.3% |
| Peer educator | 2 | 6.6% |
| Counselors | 2 | 6.6% |

**Table 3. FGD characteristics.**

| FGD | Site | N | Child age* |
|---|---|---|---|
| 1 | Bondo | 8 | 2–5 |
| 2 | Bondo | 11 | 1–2 |
| 3 | Bondo | 11 | 6–10 |
| 4 | Bondo | 6 | ≤1 |
| 5 | Ambira | 9 | 2–5 |
| 6 | Ambira | 10 | 6–10 |
| 7 | Ambira | 4 | ≤2 |
| 8 | Port Reitz | 6 | ≤5 |
| 9 | Port Reitz | 7 | 5+ |

*Targeted age categories were caregivers of children <1, 1–2, 2–5, and 6–10. However, age categories were combined at smaller sites.

## 3.1 Overview of results

Both caregivers and providers discussed how regimen-related characteristics posed challenges to caregivers and are, thus, collectively referred to throughout the results as "participants" when themes aligned. When findings differed amongst participant type, "caregiver" or "provider" is used to differentiate themes unique to each group.

Participants described how ART medications came in two primary formulations: syrups and tablets. Very young children mostly received syrups. Older children received tablets that were either crushed and dissolved in liquid for administration or as they got older, swallowed whole.

Participants discussed how regimen-related characteristics led to individual and intrapersonal challenges and impacted their family. The four primary regimen-related challenges with current formulations included: (1) poor taste, (2) complex preparation and administration, (3) required cold storage, and (4) frequent refills. Each of these regimen-related characteristics created additional challenges for the caregiver and child.

**3.1.1 Poor taste.** The poor taste of the pediatric ART was a salient theme among both caregivers and providers. Participants described how the bitterness of ART contributed to poor adherence, child refusal of the medication (running away/hiding, spitting out, vomiting) and ultimately strained the relationship between the caregiver and child. If the child vomited or spat out the ART, some caregivers discussed not knowing how much medication the child received and questioned whether they should add an additional dose.

"*The palatability is not good, and as you know, children love tasteful things and when you give them drugs that are not tasteful, they spit it out. It becomes difficult to administer the drug, so you don't even know whether the child or children have gotten the correct dosage.*" (KII_ODS_KII07)

Because of the medications poor taste, administration was often described as a "battle", with children refusing to take the medication, getting anxious around times of medication administration, and spitting out the medicine because of its poor taste. Many described how their child would run, hide, fight, or cry due to the unease around ART administration. "*It reaches within a point where if the child sees the mother taking the bottle, he/she will just run away crying that the medication is bitter and that they don't want it.*" (Hosp3_KII08).

Caregivers also described how the process of drug administration not only stressed their child but also caused them stress, anxiety, and guilt and strained the child-caregiver relationship.

*They lose control and become unhappy for a very long time to the extent that I could also shed tears because after giving the drugs the baby becomes sullen and sad which makes me question what kind of life my child is going to have. That used to disturb me a lot. It is challenging. We are giving them drugs, but it is challenging. (Hosp2_FGD2_ID04)*

To increase palatability, many caregivers mixed the solution with porridge, honey, sugar, or the syrup formulation, but "*sometimes the doctors recommend we put it in milk or porridge, and when they notice it's in the porridge, he won't take it. And when dissolved in milk, he won't even take a sip.*" (Hosp1_FGD1_ID01). Caregivers described how adding sweeteners helped ease anxiety around administration, but they could not always afford the mixers, making administration even more difficult when they were unavailable: "*She only likes tea, and it becomes hard for us, sometimes we don't have sugar, because of lack of money and the child has to take the medication so this one is usually a challenge.*" (Hosp2_FGD1_ID03).

Caregivers also described different ways in which they could enhance their child's cooperation through bribes or threats. Caregivers used instant rewards like sugar, honey, and phone time, or future rewards to motivate medication adherence: "*I have to entice the child with a promised gift. I have to promise her that after taking medication, I will buy for her French fries*" (Hosp2_FGD3_ID08). Although many caregivers utilized bribes, some discussed how bribes can be counterproductive if they cannot be maintained and provided every day, as their child would refuse to take the medication without the bribe.

"I *don't like giving mine honey. . . because when I don't have the honey and I must give the medicine, she will ask for honey. . . .*" (Hosp1_FGD2_ID06)

A handful of caregivers explained that their child responded best to threats of either physical violence or restriction of leisure activities: "*The child usually starts crying once he sees the medication so I have to scare the child by holding a cane so that they can take the medication.*" (Hosp2_FGD3_ID09). The caregivers who use threats during administration expressed an increase in their anxiety around ART but continue to do it, as it was an effective way to increase cooperation.

**3.1.2 Complex preparation and administration.** Caregivers described how the process of preparing and administering pediatric ART could take up to an hour to complete, twice a day, which placed a large burden on them. Tablets were described as the most complex to prepare, requiring them to first cut and crush the tablet then measure, dissolve, and mix the formulation. Each of these steps introduced opportunities for error.

"*I prepare porridge and then crush the medication then I take porridge and pour on the bottle cup and add the medication I had crushed, once I do this, I add porridge on top then give the child, I divide the medication into two, I give the first half first then the rest again with porridge, because when taken like that it is too bitter, so that is what I do.*" (Hosp3_FGD1_ID06).

When a child was too small to take a full dose, caregivers were required to cut the tablet to give the proper dose. Although the tablets have marks indicating where the tablet needed to be broken, it did not always break in the proper spot, forcing the caregiver to choose whether to underdose or overdose because "*Sometimes it breaks wrongly, forcing you to give ¾ instead of a ½ the tablet and . . .You just give it like that.*" (Hosp1_FGD1_ID03)

After measuring the proper dose, the tablet was crushed and dissolved in water. However, tablets were described as difficult to crush and did not dissolve well, leaving the mixture sticky and layered: "*It is hard for him to swallow and when chewing it sticks on his teeth. Sometimes he even spits the tablet. (Hosp1_FGD1_ID04)*"

Given it's complexity both caregivers and providers preferred medication administration to be done by the same person each day. Occasionally caregivers described how they must delegate the task to a trusted family member, friend, or older sibling that was aware of the child's HIV status. They explained how the complexity of medication preparation and administration led them to "*think that they might not give the drug well, so I have to call and confirm that the child has been given medication. (Hosp2_FGD3_ID09)*" Providers specifically noted that children living with HIV with primary caregivers who rely heavily on secondary caregivers, tend to have higher viral loads at checkups. This can be due to knowledge gaps, forgetfulness, and complexity of administration: "*knowledge gap is there since the grandmother is not that learned so maybe sometimes reading the instructions is not easy for them, and they also have other priorities to attend to (Hosp2_KII04).*"

Caregivers across all focus groups expressed a strong desire to keep their child's HIV status private from those in the community, only disclosing the status to close friends and family, due to the stigma surrounding HIV. Fear of disclosure due to complex medication administration process made traveling difficult for children with HIV and their caregivers, as their caregivers struggle with how to carry the medication and properly prepare and administer it discreetly. Caregivers explained that traveling regularly made them question whether they should delay medication administration or risk disclosing their child's status.

*"When there is a funeral at home and you attend it, when it comes time to give the child medication and there are some relatives who are unaware of your status, when you take out the medication to give to the child, they ask a lot of questions. It forces us to skip giving the child their medication to avoid the questions." (Hosp2_FGD2_ID03)*

Furthermore, this complex preparation and administration sometimes interfered with daily schedules, especially in situations where caregivers worked early mornings or late evening or if the children were in school. In the morning, the difficulty came if they needed to re-administer the medication if the child vomited, and in the evening, the difficulty was with getting home at the proper time to administer the medication.

*"I also had a challenge with the timing, especially if the child is going to school…you find that to administer the drugs, the child needs to eat, and sometimes the child will vomit after taking the drug… where you'll have to wait for a while, before re-administering the drug…. and maybe you are running late." (Hosp1_FGD2_ID01)*

Together, the need to administer medications twice daily, the complex process of preparation and administration, and the "battles" that occurred at medication administration times, presented multiple opportunities for participant's HIV status to be discovered by neighbors and family members. Caregivers explained many ways ART administration can inadvertently disclose their child's status including: the sound of the pill bottles rattling in their purse, traveling monthly to refill medication, not allowing visitors in the house in the evenings, and having the child repeatedly question why they need daily medication.

*"So, when I want to give the baby the drugs, I always like to be alone and when I hear footsteps approaching; I'd want to know who it is. I just tell them I'm a bit busy and tell them to wait a while until I'm done. Personally, I'm not scared but this is my child's life and her privacy." (Hosp2_FGD2_ID04).*

As caregivers begin the process of HIV disclosure to their child, either in part or in whole, it allowed the children to take part in their health and hold some responsibility. Once children understood the importance of the medication, the caregivers explained how it aided in the process of administration, even if their children still refused to swallow tablets whole.

*I told mine that his body is invaded by viruses, and the drug is meant to make them inactive. If he does not take his medication, the virus will become active and affect him. He is well informed and aware that he can get sick if he does not adhere to his medication. (Hosp1_FGD2_ID03)*

Caregivers with older children explained how the process of administration got easier with age. Not only were the challenges of taste and cooperation were minimized among older

children who were able to swallow pills whole, but caregivers explained the children as young as six or seven had begun taking responsibility for their own or their sibling's medication adherence.

> *Since one is 7 years old and can read, she is even able to administer the medication to the younger sibling, as she takes hers.*

**3.1.3 Storage.** Of the medications discussed, all required a storage environment that was cool, dry, and away from direct sunlight, with the syrups specifically requiring refrigeration. Providers discussed how improper storage of the medication led to a decreased efficacy of the drug. "*So actually, that has led to high viral loads because they are taking medication that not. . . that is not well stored. The condition has been compromised, the quality.*" (Hosp2_KII08).

When caregivers did not have access to a refrigerator in their home, they described how they improvised cool areas to store the medication. Caregivers stored the medication in a cool jug of water or in a sandy hole in the ground. Each of these storage methods required frequent attention to ensure the medication environment does not get too warm and will not be discovered by young children or visitors.

> "*I took a tin of sand and water and kept in in a cool place in my bedroom, so once I had sealed the bottle tightly, I would insert the bottle in the water, so I made sure that I kept on changing the water so that it does not become warm.*" (Hosp3_FGD2_ID03)

The strict storage environments also made it difficult in times of travel, as these strategies were not discreet and impossible to maintain on public transport. Tablets, however, could be kept in drawers, cupboards, or even in a handbag while traveling.

**3.1.4 ART refill frequency and stockouts.** The frequency of ART refill varied depending on child age and current viral load, because providers utilize medication refills as checkups to monitor weight-based dosing and medication adherence. Younger children and those with high viral loads had 2-week refill appointments, while the others varied from one to two months between appointments. Providers utilized refill appointments to check adherence by pill counts. Providers emphasized that an individualistic approach to each situation was important in determining the frequency of ART refill. The frequent refills posed multiple challenges for caregivers, as they had work commitments and responsibilities outside of the home.

> "*There was a time that every month, my two girls had to miss their classes because of going for the refills, but it reached a point where I started skipping going with them to the clinic. Whenever they ask, I would tell them that, they were sitting for exams. . .I came all alone for about 5 months. You know, it even aroused my neighbors' curiosity, they would always ask; "What's the matter that makes you take the children to the hospital every month?"* (Hosp1_FGD1_ID04).

Many caregivers struggled to get fare required for transport, getting time off from work, and making their child miss school. The caregivers felt obligated to justify why their child was missing school and why they were missing work: "*Also, it is very challenging to the school going kids. You will lie to the teacher today, but what will you tell him tomorrow?*" (Hosp1_FGD1_ID02).

Providers described how supply stockouts or shortages exacerbated the complexity of ART administration. When a pediatric drug stockout would occur, the providers were forced to

hand out adult versions of the medication, which led to prescribing large, bitter pills that were difficult to accurately dose. Supply shortages would require caregivers to (1) increase frequency of trips to the hospital to collect smaller quantities of drugs, (2) seek drugs from private institutions at a cost to them, or (3) skip doses.

> "*And the challenge we have now . . . there is low supply or there are no pellets around. What we have remaining in form of lopinavir is now the bigger tablets for the adults. In that situation we are stuck on what to do. Because you cannot resolve to maybe going to the bigger tablets and breaking it. You won't know the exact quantity that you have given them. So, supplies are a challenge.*" (Hosp3_KII06).

In times of supply shortages, children could be prescribed a syrup and tablet at the same time, or to be switched from one to the next at refill appointments. Providers noted that a child on multiple formulations led to confusion and mix-ups on administration. They emphasized that caregivers did the best they can with the time/skill they have, but improper handling of the drugs still led to suboptimal outcomes. For example, certain tablets cannot be crushed, but "*here is a situation whereby there is a dilemma between giving a whole tablet and the child will refuse to swallow completely and it would go to waste or you crush it and compromise the effectiveness of drug and at least deliver something to this child* (Hosp3_KII10)."

**3.1.5 Challenges to adherence: In context.** The many challenges discussed above were accentuated by financial constraints, religion, educational background, work status, and responsibilities outside the home that affected the caregiver's ability to appropriately administer ART: "*Most of these clients you find that from where they come from, they are not financially stable (KII_ODS_Bondo_02).*" It was not uncommon for the caregivers to have scarce resources and be forced to choose between whether to spend money on transportation and medication or food: *Sometimes the caregiver doesn't have money to come to the facility and completing tasks of the caregivers. (Hosp2_KII05).*

The topic of access to information, education, counseling, and knowledge influenced provision of pediatric ART was a common occurrence in the provider interviews, but not as salient in the caregiver focus group discussions. Providers documented the challenge with proper measuring and administration of medication via primary or secondary caretakers who were illiterate or had a low education level: "*I would not say illiterate for lack of a better term, but the level of education determines if they dispense the right volume, so maybe the child is overdosed or underdosed because of lack of knowledge on how to titrate* (Hosp3_KII03)."

In addition to challenges with administration, a lack of knowledge about HIV and opposing religious beliefs made it difficult to convince caregivers that their child requires medication. Providers mentioned that these caregivers can go through a denial stage where they do not believe their child will become sick from HIV or need medication until opportunistic diseases takeover: "*Some of them will tell you that their church does not accept use of medication, so the child and mother will not use the medication for a long time, so if they notice they have been overwhelmed by opportunistic diseases, that is when the lady or the child is brought back to the clinic, then you start again from there." (*Hosp3_KII01*).*

## 4.0 Discussion

Studies have shown that approximately 50% of Kenyan pediatric patients on ART have interruptions in their treatment lasting greater than 48 hours [13], with 48% being lost to follow up within 5 years [14]. This study explored the challenges that current standard of care pediatric ART administration presents to caregivers, children, and healthcare providers. The testimony

from the stakeholders (caregivers and providers) highlighted the emotional toll that adhering to life-saving medication can have on both caregivers and children.

Long term medication adherence is a complex dynamic influenced by both the caregiver's approach and the child's acceptance and is a process that shifts as the child ages [15]. While studies have shown that greater caregiver involvement in medication administration leads to better adherence outcomes, as the child ages some responsible for medication is shifted to the child. Many programs aimed at transitioning children to adult care start at 12, 15, 18 years of age or even older [16]. Even amongst older children, readiness to take on responsibility for medication adherence is dependent on several factors including personal characteristics and mental health of the child, family structure, pre-transition adherence, disclosure of HIV status, and the caregiver must be able to assess these factors accurately when beginning to allow the child to take responsibility. Our study–and others–suggest that this shift may begin much earlier, with children as young as 6 or 7 beginning to take responsibility for their own medication adherence. Simplification of medications will help improve treatment adherence among children who may need to take on this responsibility before they are developmentally ready [17]. However, the need for caregiver support remains critical at every age and there is a strong link to positive caregiver involvement and their child's adherence [15, 18, 19].

The complexity of ART administration was a salient theme in our results, with caregivers and providers lamenting the complex process of measuring and administering syrups or crushing tablets and mixing with food or drink multiple times per day. Simplifying ART regimens can improve adherence in both children and adults [20–23]. ART regimens can be simplified through reducing the number of doses required per day and minimizing the pill burden or liquid volume associated with a single dose to improve adherence. In the past several years, several new delivery mechanisms have been explored and developed to ease the burden of administration. These include nanoparticle and dispersible tablets that can be mixed with food and beverage that studies have shown to be effective and acceptable, especially to younger children [24]. In addition, new promising advances in pediatric ART may improve long term adherence among children and alleviate the strain on the relationship between caretakers and children living with HIV. These advances include and buccal films [25], long-acting subcutaneous, intramuscular, or implant formulations and microarray patches [26–30]. While these long-acting formulations may overcome many of the challenges associated with current formulations, they may introduce other challenges including inability to adjust to frequent weight-based dosing changes that occur in early childhood, pain and irritation at the injection site, and potential for stigma due to large patches [31].

Our team is developing oral dissolvable strips (ODS) [32] containing WHO recommended regimens for infants and children that feature several characteristics that reduce barriers to ART adherence: simplified drug preparation and administration, easily adaptable dosing to accommodate infant growth, and discreet packaging. The strip will be ready-to-use: caregivers adhere an ODS to the palate, inner cheek, or tongue for rapid dissolution in saliva. In pre-clinical studies, ODS formulated with prophylactic and therapeutics doses of ARV showed in-vitro and in-vivo bioequivalence to standard formulations [33]. Due to the current barriers, providers and caregivers were enthusiastic about being a part of the conversation to brainstorm better ways to administer ART. The results of this study reinforce the importance of creating a novel, user-centered design for pediatric ART to increase adherence, decrease stress, and overall improve the quality of life of both patients and caregivers treating HIV. This paper is part one of a three-part study that aims to take information from these FGDs and provider interviews to create prototypes of oral dissolvable strips (ODS) for pediatric ART treatment. In future steps, the team will develop ODS and packaging prototypes and present to stakeholders (caregivers, children, and providers) for feedback and finalization of design element.

A few limitations should be noted. The participants in the FGDs were caregivers who were actively engaged in their child's ART care and were willing to participate in a focus group discussion regarding HIV, thus, had disclosed to someone. These caregivers may not represent the larger community, including those who are unengaged in their child's HIV care, who remain undisclosed, and who are not comfortable speaking about their child's HIV status with others. Thus, implications for pediatric adherence among study participants may be more favorable than the normal. Although this study focused on the multi-level challenges that administration of pediatric HIV ART presents to caregivers and health systems, many of the individual level challenges can be translated to other forms of chronic illness and their treatment plans.

## 5.0 Conclusion

These findings highlight the many challenges that caregivers face in providing ART to their childing living with HIV and highlight the need for improved formulations for pediatric ART to ease the daily burden on caregivers and children to increase adherence, improve child health, and overall quality of life of families.

## Acknowledgments

We would like to thank the caregivers and providers who participated in the discussion and without whom these results would not be possible.

## Author Contributions

**Conceptualization:** May Maloba, Catherine Wexler, Edward Maliski, Gregory Thomas, Nicodemus Maosa, Sarah Finocchario-Kessler.

**Data curation:** May Maloba, Shadrack Kale.

**Formal analysis:** Michala Sliefert, Catherine Wexler, Sarah Finocchario-Kessler.

**Funding acquisition:** Edward Maliski, Sarah Finocchario-Kessler.

**Investigation:** May Maloba, Edward Maliski, Zachary Nicolay, Gregory Thomas, Sarah Finocchario-Kessler.

**Methodology:** May Maloba, Frederick Were, Yvonne Mbithi, George Mugendi, Edward Maliski, Zachary Nicolay, Gregory Thomas, Sarah Finocchario-Kessler.

**Project administration:** May Maloba, Catherine Wexler, Frederick Were, Zachary Nicolay, Shadrack Kale, Nicodemus Maosa, Sarah Finocchario-Kessler.

**Supervision:** Sarah Finocchario-Kessler.

**Writing – original draft:** Michala Sliefert, Catherine Wexler.

**Writing – review & editing:** Michala Sliefert, May Maloba, Catherine Wexler, Frederick Were, Yvonne Mbithi, George Mugendi, Edward Maliski, Zachary Nicolay, Gregory Thomas, Shadrack Kale, Nicodemus Maosa, Sarah Finocchario-Kessler.

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
