## [Decision Letter · Decision Letter 0]

19 Jun 2023

PONE-D-23-14403Challenges with pediatric antiretroviral therapy administration: qualitative perspectives from caregivers and HIV providers in KenyaPLOS ONE

Dear Dr. Wexler,

Thank you for submitting your manuscript to PLOS ONE. After careful consideration, we feel that it has merit but does not fully meet PLOS ONE’s publication criteria as it currently stands. Therefore, we invite you to submit a revised version of the manuscript that addresses the points raised during the review process.

I emphasis making the following revisions, while attending to all the comments from both Reviewers, which I find valid:

Engage more relevant literature on the topic, restructure the results section and place the results within the context of previous work in this area.

Also, undertake detailed discussion of the findings, and not a repetition of the results. Further, thoroughly edit the grammar. Also, determine and specify who is a child, and who is not, in this study, and determine if this will change the title of your work, especially with reference to the word ‘pediatric’.

Thank you.

Academic Editor

We look forward to receiving your revised manuscript.

Kind regards,

Adobea Yaa Owusu, PhD MPH

Academic Editor

PLOS ONE

Journal Requirements:

"We would like to thank the caregivers and providers who participated in the discussion and without whom these results would not be possible. This study was funded through the National Institute of Child Health and Human Development, Award R21HD105534."

"This study was funded by the National Institute of Child Health and Human Development, award number R21HD105534 awarded to Dr. Sarah Finocchario-Kessler. The funders had no role in study design, data collection and analysis, decision to publish, or preparation of the manuscript."

Additional Editor Comments:

Dear Dr. Catherine Wexler,

I emphasis making the following revisions, while attending to all the comments from both Reviewers, which I find valid:

Engage more relevant literature on the topic, restructure the results section and place the results within the context of previous work in this area.

Also, undertake detailed discussion of the findings, and not a repetition of the results. Further, thoroughly edit the grammar. Also, determine and specify who is a child, and who is not, in this study, and determine if this will change the title of your work, especially with reference to the word ‘pediatric’.

Thank you.

Academic Editor

Reviewers' comments:

Reviewer's Responses to Questions

**Comments to the Author**

1. Is the manuscript technically sound, and do the data support the conclusions?

Reviewer #1: Yes

Reviewer #2: Partly

2. Has the statistical analysis been performed appropriately and rigorously? 

Reviewer #1: N/A

Reviewer #2: N/A

3. Have the authors made all data underlying the findings in their manuscript fully available?

Reviewer #1: No

Reviewer #2: No

4. Is the manuscript presented in an intelligible fashion and written in standard English?

Reviewer #1: Yes

Reviewer #2: No

5. Review Comments to the Author

Reviewer #1: General Comment

This article explores an important area and provides insights into the challenges around ART Administration among children from caregivers and providers perspective. I found this to be a highly informative article with a rich results section. The article will benefit from restructuring of the results section and placing the results within the context of previous work in this area.

Specific comments

1. The authors should include where the study was conducted in the abstract and the methods section. This is missing and only appears in table 2, elsewhere the location is referred to as “three hospitals in Kenya” which is too general.

2. The methods section misses detail around how the study was conducted? Was the process iterative? Were the FGD guides refined between different FGDs? How many analysts coded?

3. For how many children had disclosure been done? This would be an important addition to the results section.

4. Table 2 would benefit from a clear description on the legend to help readers understand it.

5. The structure and classification in the results section is hard to follow. Authors need to refine this for clarity and better flow.

6. The discussion is plain and lacks detail. I worry that the article does not use the opportunity well to show the intersection of their results with prior work done in the same area. I notice articles that I would expect to be cited missing in this manuscript. The references in the article are much fewer that I would expect for this body of work. The authors should review available literature and include these to enrich their discussion. In the current state the discussion largely feels like a repetition of the results.

Reviewer #2: Thank you for an interesting manuscript that focuses on an important topic and provides an opportunity for caregivers and healthcare providers to share their perspectives. In my opinion, the manuscript is based on a study that was ethically considered, implemented, and reported. However, I have a few queries and suggestions for improvement of the manuscript.

Abstract

Results: Highlight what was reported by both groups (caregivers, healthcare providers) and then what was unique to each group.

Introduction

This section is well written but consider adding a paragraph that focuses on the current pediatric ART regimens in use in Kenya, and especially the region where you collected your data.

Comment: ‘In some cases, this also requires boiling water to make it potable, adding to the burden of ART preparation and administration for children.’ In addition, many Kenyan families struggle to get both water and fuel (e.g. kerosene, charcoal, firewood, gas).

Methods

Edit grammar here – ‘Participants were eligible to participate in focus group discussions if cared for child living with HIV…’

‘Caregivers included mothers, fathers, grandparents, or other family members serving as primary caregivers…’ Who are these other family members? Do you have this data? If yes, be explicit.

Consider using the COREQ checklist or any similar tool to help you report your study methodology fully. You are missing details and I will flag them as much as possible.

‘Caregivers were recruited … or providers with established relationships with clients at the study sites.’ What does this (established relationships with clients) mean?

‘Participants were eligible to participate in focus group discussions if cared … at one of the study hospitals.’

-You need to include information on the type of health facilities you used

-You need to report the number of facilities in the methodology chapter

-Justify use of the three facilities. Why 3 only? Why these particular 3 facilities?

‘Providers were eligible for interviews if they routinely participated in pediatric HIV care…’

-Did it matter how long they had been serving at the clinic or any other offering similar care?

-Can you add more information on the cadres of healthcare workers that were eligible?

‘Participants provided written informed consent in compliance with IRB approvals… for human subjects’ research.’ In this and all future research, consider using ‘human participants’ as opposed to ‘human subjects.’ The word ‘subject’ has over time been dropped in many settings because of its historical connotations and dehumanizing and disrespectful quality.

‘Immediately after informed consent was provided…’

-Add here details on the informed consent process. Step by step. Who did it? Where was it done? How was it done? And so on.

- Your study used other languages besides English. Add information on informed consent, translation and associated documents

Edit grammar here – ‘…surveys with each participant were conducted to capture socio-demographics characteristics or providers and caregivers.’

‘Surveys occurred in a private room and were conducted by a research assistant in English, Swahili, or Luo.’ This study has KIIs and FGDs. It is not clear what you are referring to here when you describe surveys.

Add all detail on the FGDs and KIIs. For example (and not conclusive):

-Where –venue- exactly were they done (and why there)?

-For each FGD, provide details on participants e.g. how many?

-For each FGD, how many facilitators were involved?

-Did you need to match facilitators to participant characteristics?

-Did you serve any refreshments?

-Did you provide any reimbursements to the participants?

‘Focus groups and interviews were audio recorded and transcribed verbatim.’ Your study used other languages besides English. Add information on translation?

‘Analysts met periodically to refine the codebook and develop consensus…’

-How many coders were involved?

‘Focus groups and KIIs lasted approximately 90 or 30 minutes, respectively.’

Move this sentence to the data collection section. It does not fit in the data analysis bit.

Results

‘The healthcare providers included… and others’

-Who are these in the category ‘other’? Be explicit.

‘About 76% of the caregivers were living with HIV.’

In Table 1, is ‘disclosure’ referring to disclosure about the child’s or the caregiver’s status? Clarify.

‘…who provided care for n=83 children (mean age 4.7 years old, range 5 months-18 years old)…’ Pg 9. 18-year-olds are defined as adults in Kenya.

In some sections, it is not clear whether the findings reported are from caregivers or healthcare providers. Ensure you do a thorough analysis of your findings. Presenting codes and themes is not adequate. Did findings differ with participant characteristics? Category of participants?

‘Types of Pediatric ART Regimens:

Participants described how ART medications…’

Which category of participants are you referring to here? Qualify. This applies in all sections of the results chapter where ‘participants’ are referred to generally.

‘Most caregivers did not have access to a refrigerator…’

Where is this type of data coming from?

‘Many described how their child would run…’

What is this referring to? A percentage of the FGDs?

Why is ‘poor taste’ discussed in two different sections? This is confusing. Consider exhaustively discussing one theme and then proceeding to the next.

‘Transcripts were coded based on a priori themes related to 1) individual-level considerations (e.g.,

knowledge and beliefs), 2) community-level considerations (e.g., privacy/confidentiality), 3) process of dosing (e.g., child refusal), and 4) environmental/system level considerations (e.g., frequency of medication refill).’ Pg 9.

Organize the findings chapter using these four distinct themes.

Some quotes are not in italics e.g. pg 12. Be consistent.

Edit grammar in this chapter e.g. ‘The topic of how access to information …’

Discussion

Ensure this chapter only discusses the findings from the results chapter.

Consider reporting them in four paragraphs guided by the reorganization of the results: 1) individual-level considerations, 2) community-level considerations, 3) process of, and 4) environmental/system level considerations.

‘We primarily focused on caregivers with young children living with HIV who required more complex administration with syrups or crushed tablets as their primary ART formulation.’

On Pg 9, you report that the children included 18-year-olds.

‘This paper is part one of a three-part study that aims to take information from these FGDs and provider interviews to create prototypes of oral dissolvable strips (ODS) for pediatric ART treatment…’

This information is needed at the end of the introduction

6. PLOS authors have the option to publish the peer review history of their article (what does this mean?). If published, this will include your full peer review and any attached files.

Reviewer #1: No

Reviewer #2: **Yes: **Violet Naanyu

---

## [Author Response · Author response to Decision Letter 0]

29 Aug 2023

Dear reviewers and editor, 

We appreciate your review of our manuscript and your extremely helpful comments to improve the clarity of the methods and the contextualization of the results. Per your recommendations, we have added details to the methods, tried to improve the organization of the results, and added details in the discussion to help contextualize our study within the broader literature around this topic. We believe these changes have significantly improved our manuscript and appreciate the time and effort that reviewers dedicated to their review. 

A point-by-point response is included below. 

Editor Comments

We have included the “inclusivity in global research” questionnaire in our submission.

"We would like to thank the caregivers and providers who participated in the discussion and without whom these results would not be possible. This study was funded through the National Institute of Child Health and Human Development, Award R21HD105534." We note that you have provided funding information that is not currently declared in your Funding Statement. However, funding information should not appear in the Acknowledgments section or other areas of your manuscript. We will only publish funding information present in the Funding Statement section of the online submission form. 

Please remove any funding-related text from the manuscript and let us know how you would like to update your Funding Statement. Currently, your Funding Statement reads as follows: "This study was funded by the National Institute of Child Health and Human Development, award number R21HD105534 awarded to Dr. Sarah Finocchario-Kessler. The funders had no role in study design, data collection and analysis, decision to publish, or preparation of the manuscript."

We have removed funding information from the acknowledgements. The funding statement entered into the submission system is correct.

4. Engage more relevant literature on the topic, restructure the results section and place the results within the context of previous work in this area.

We have reorganized the results and added relevant literature and content to the discussion section to contextualize our finding more thoroughly within recent literature.

5. Also, undertake detailed discussion of the findings, and not a repetition of the results. 

We have provided a more detailed discussion.

6. Further, thoroughly edit the grammar. 

We have edited grammar.

7. Also, determine and specify who is a child, and who is not, in this study, and determine if this will change the title of your work, especially with reference to the word ‘pediatric’.

While we recognize that people 18 and older are legally adults, the caregiver to the 18-year-old “child” in this study, also cared for two younger children living with HIV: 9 months and 3 years; and, thus, met the eligibility criteria of the study. This grandparent talked about all three of her grandchildren during the survey and interview and, thus, we included the 18-year-old in the summary of caregiver characteristics. We have clarified this in the text. Given our focus on pediatric ART, we do not believe that this single older “child” necessitates a title change.

Reviewer 1

1. The authors should include where the study was conducted in the abstract and the methods section. This is missing and only appears in table 2, elsewhere the location is referred to as “three hospitals in Kenya” which is too general.

We have added the study hospitals in the main text, as well as the represented counties within the abstract.

2. The methods section misses detail around how the study was conducted? Was the process iterative? Were the FGD guides refined between different FGDs? How many analysts coded?

We have added details, per the CORE-Q checklist, to the methods section of the manuscript.

3. For how many children had disclosure been done? This would be an important addition to the results section.

We have added in that 20 out of the 83 children knew their HIV status.

4. Table 2 would benefit from a clear description on the legend to help readers understand it.

We have reformatted the tables to be clearer and easier to understand.

5. The structure and classification in the results section is hard to follow. Authors need to refine this for clarity and better flow.

We have re-organized the results to emphasize how regimen-related challenges create challenges in other dimensions of the participant’s life. The organization is briefly summarized in the “overview of results” section to justify this organization and orient the reader. We have also removed some of the subheadings which may have complicated the results. We hope this organization is easier to follow. 

6. The discussion is plain and lacks detail. I worry that the article does not use the opportunity well to show the intersection of their results with prior work done in the same area. I notice articles that I would expect to be cited missing in this manuscript. The references in the article are much fewer that I would expect for this body of work. The authors should review available literature and include these to enrich their discussion. In the current state the discussion largely feels like a repetition of the results.

We have added content to the discussion to represent the current literature more thoroughly around pediatric ART and recent advances.

Reviewer #2: 

Thank you for an interesting manuscript that focuses on an important topic and provides an opportunity for caregivers and healthcare providers to share their perspectives. In my opinion, the manuscript is based on a study that was ethically considered, implemented, and reported. However, I have a few queries and suggestions for improvement of the manuscript.

1. Results: Highlight what was reported by both groups (caregivers, healthcare providers) and then what was unique to each group.

Throughout the results we use the term “participant” when both caregivers and providers noted that theme; while the terms “caregiver” or “provider” is used to describe themes unique to each group. We have added a sentence at the beginning of the results to clarify this structure.

Introduction: 

2. This section is well written but consider adding a paragraph that focuses on the current pediatric ART regimens in use in Kenya, and especially the region where you collected your data.

We have added a line in the intro outlining current first line ART, as well as availability of these ART as liquid or tablet formulations.

3. Comment: ‘In some cases, this also requires boiling water to make it potable, adding to the burden of ART preparation and administration for children.’ In addition, many Kenyan families struggle to get both water and fuel (e.g. kerosene, charcoal, firewood, gas).

We have added fuel costs as an additional burden.

Methods: 

4. Edit grammar here – ‘Participants were eligible to participate in focus group discussions if cared for child living with HIV…’

Edited. 

5. ‘Caregivers included mothers, fathers, grandparents, or other family members serving as primary caregivers…’ Who are these other family members? Do you have this data? If yes, be explicit.

This information is reported in Table 1.

6. Consider using the COREQ checklist or any similar tool to help you report your study methodology fully. You are missing details and I will flag them as much as possible.

We have modified the methods to include the COREQ checklist items and have included this in the resubmission packet.

7. ‘Caregivers were recruited … or providers with established relationships with clients at the study sites.’ What does this (established relationships with clients) mean?

We have specified that mentor mothers and providers who identified and recruited participants were part of the client’s clinical care team.

8. ‘Participants were eligible to participate in focus group discussions if cared … at one of the study hospitals.’

a. -You need to include information on the type of health facilities you used

b. -You need to report the number of facilities in the methodology chapter

c. -Justify use of the three facilities. Why 3 only? Why these particular 3 facilities?

We have added this information. 

9. Providers were eligible for interviews if they routinely participated in pediatric HIV care…’

a. -Did it matter how long they had been serving at the clinic or any other offering similar care?

b. -Can you add more information on the cadres of healthcare workers that were eligible?

This has been included in new Table 2.

10. ‘Participants provided written informed consent in compliance with IRB approvals… for human subjects’ research.’ In this and all future research, consider using ‘human participants’ as opposed to ‘human subjects.’ The word ‘subject’ has over time been dropped in many settings because of its historical connotations and dehumanizing and disrespectful quality.

We have dropped this phrase from that sentence.

11. ‘Immediately after informed consent was provided…’

a. -Add here details on the informed consent process. Step by step. Who did it? Where was it done? How was it done? And so on.

b. - Your study used other languages besides English. Add information on informed consent, translation and associated documents

This information has been added.

12. Edit grammar here – ‘…surveys with each participant were conducted to capture socio-demographics characteristics or providers and caregivers.’

This has been corrected.

13. ‘Surveys occurred in a private room and were conducted by a research assistant in English, Swahili, or Luo.’ This study has KIIs and FGDs. It is not clear what you are referring to here when you describe surveys.

Per the sentence above, “surveys with each participant were conducted to capture socio-demographics characteristics of providers and caregivers.” 

14. Add all detail on the FGDs and KIIs. For example (and not conclusive):

a. -Where –venue- exactly were they done (and why there)?

b. -For each FGD, provide details on participants e.g. how many?

c. -For each FGD, how many facilitators were involved?

d. -Did you need to match facilitators to participant characteristics?

e. -Did you serve any refreshments?

f. -Did you provide any reimbursements to the participants?

We have added this information into the methods section.

15. ‘Focus groups and interviews were audio recorded and transcribed verbatim.’ Your study used other languages besides English. Add information on translation?

We have added information on translation.

16. ‘Analysts met periodically to refine the codebook and develop consensus…’

a. -How many coders were involved?

We have added this information.

17. ‘Focus groups and KIIs lasted approximately 90 or 30 minutes, respectively.

a. Move this sentence to the data collection section. It does not fit in the data analysis bit.

This has been moved.

Results

18. ‘The healthcare providers included… and others’: Who are these in the category ‘other’? Be explicit.

We have included this information in new Table 2.

19. ‘About 76% of the caregivers were living with HIV: In Table 1, is ‘disclosure’ referring to disclosure about the child’s or the caregiver’s status? Clarify.

We have clarified within the table that it is referred to disclosure of the child’s HIV status to anyone but the caregiver involved in the FGD.

20. …who provided care for n=83 children (mean age 4.7 years old, range 5 months-18 years old)…’ Pg 9. 18-year-olds are defined as adults in Kenya.

While we recognize that people 18 and older are legally adults, the caregiver to the 18-year-old “child” in this study, also cared for two younger children living with HIV: 9 months and 3 years; and, thus, met the eligibility criteria of the study. This grandparent talked about all three of her grandchildren during the survey and interview and, thus, we included the 18-year-old in the summary of caregiver characteristics. We have clarified this in the text.

21. In some sections, it is not clear whether the findings reported are from caregivers or healthcare providers. Ensure you do a thorough analysis of your findings. Presenting codes and themes is not adequate. Did findings differ with participant characteristics? Category of participants?

There was significant alignment between caregiver and provider themes; thus, we presented many of them together to avoid duplication. Throughout the results we use the term “participant” when both caregivers and providers noted that theme; while “caregiver” or “provider” is used to describe themes unique to each group. We have added a sentence at the beginning of the results to clarify this structure.

22. ‘Types of Pediatric ART Regimens: Participants described how ART medications…’

Which category of participants are you referring to here? Qualify. This applies in all sections of the results chapter where ‘participants’ are referred to generally.

See response to query 21.

23. ‘Most caregivers did not have access to a refrigerator…’ Where is this type of data coming from?

We slightly modified this sentence because we do not have an exact count of the number with or without refrigerators. However, the theme of challenges with cool storage outside of refrigerator was mentioned 32 times by 14 participants. Of these, only one participant whose child’s medication needed to be kept cool spoke of owning a refrigerator, while others discussed keeping in jugs of cool water or using sand to keep the syrups cool.

24. ‘Many described how their child would run…’ What is this referring to? A percentage of the FGDs?

Child unease around medication administration was a strong theme, mentioned 32 times by caregivers in 8 of the 9 focus group discussions. 

25. Why is ‘poor taste’ discussed in two different sections? This is confusing. Consider exhaustively discussing one theme and then proceeding to the next.

We have reorganized the results to remove headings and emphasize how regimen-related challenges created challenges in other aspects of the participant’s life. 

26. ‘Transcripts were coded based on a priori themes related to 1) individual-level considerations (e.g.,

knowledge and beliefs), 2) community-level considerations (e.g., privacy/confidentiality), 3) process of dosing (e.g., child refusal), and 4) environmental/system level considerations (e.g., frequency of medication refill).’ Pg 9.: Organize the findings chapter using these four distinct themes.

We had initially done this; however, re-organized prior to initial submission because there was so much overlap between these various levels that we found ourselves repeating themes multiple times. For example, community level considerations (privacy, confidentiality, unintentional disclosure) were exacerbated significantly because children would run, cry and hide from the process of administration and this process of administration was so challenging because the drugs were so bitter, which is regimen-related characteristic. We found that while we may have coded that way, there was no simple and straight-forward way to separate these distinct levels as we discuss the broader context of pediatric ART administration. Thus, we have re-organized to emphasize how regimen-related challenges create challenges in other dimensions of the participant’s life. We have tried to simplify this organization and prep the reader more for it in our results section.

27. Some quotes are not in italics e.g. pg 12. Be consistent.

We’ve corrected this.

28. Edit grammar in this chapter e.g. ‘The topic of how access to information …’

This has been corrected.

Discussion

29. Ensure this chapter only discusses the findings from the results chapter.

We have gone through and reorganized the results and discussion to provide a more in-depth discussion of the implications of this research.

30. Consider reporting them in four paragraphs guided by the reorganization of the results: 1) individual-level considerations, 2) community-level considerations, 3) process of, and 4) environmental/system level considerations.

Please see response to comment #26 for reasoning behind our chosen organization.

31. ‘We primarily focused on caregivers with young children living with HIV who required more complex administration with syrups or crushed tablets as their primary ART formulation.’

On Pg 9, you report that the children included 18-year-olds.

We have clarified that the caregiver of the 18-year old also cared for young children. While the 18-year old was not inquired upon, the caregiver did include them as a child she cared for and, thus, we reported here.

32. ‘This paper is part one of a three-part study that aims to take information from these FGDs and provider interviews to create prototypes of oral dissolvable strips (ODS) for pediatric ART treatment…’ This information is needed at the end of the introduction

We have added this in the intro.

---

## [Decision Letter · Decision Letter 1]

15 Nov 2023

PONE-D-23-14403R1Challenges with pediatric antiretroviral therapy administration: qualitative perspectives from caregivers and HIV providers in KenyaPLOS ONE

Dear Dr. Wexler,

Thank you for submitting your manuscript to PLOS ONE. After careful consideration, we feel that it has merit but does not fully meet PLOS ONE’s publication criteria as it currently stands. Therefore, we invite you to submit a revised version of the manuscript that addresses the points raised during the review process.

We look forward to receiving your revised manuscript.

Kind regards,

Petros Isaakidis

Academic Editor

PLOS ONE

Reviewers' comments:

Reviewer's Responses to Questions

**Comments to the Author**

1. If the authors have adequately addressed your comments raised in a previous round of review and you feel that this manuscript is now acceptable for publication, you may indicate that here to bypass the “Comments to the Author” section, enter your conflict of interest statement in the “Confidential to Editor” section, and submit your "Accept" recommendation.

Reviewer #2: (No Response)

2. Is the manuscript technically sound, and do the data support the conclusions?

Reviewer #2: Partly

3. Has the statistical analysis been performed appropriately and rigorously? 

Reviewer #2: N/A

4. Have the authors made all data underlying the findings in their manuscript fully available?

Reviewer #2: (No Response)

5. Is the manuscript presented in an intelligible fashion and written in standard English?

Reviewer #2: Yes

6. Review Comments to the Author

Reviewer #2: ABSTRACT

-In your abstract, as you report findings, highlight what was reported by both groups (caregivers, healthcare providers) and then what was unique to each group. This is still pending.

-Consider having a conclusion, not a discussion section in the abstract.

METHODS

-Your study used other languages besides English. Add information on informed consent, translation and associated consenting documents. This is still pending.

-It is still not clear which findings come from which of the 9 FGDs and/or which of the KIIs. I suggest you get 3 index refereed articles to guide you in your data analyses and reporting. You need to:

*Run analysis for the 9 FGDs

*Run analysis for each of the 30 KIIs

*Run a summary that shows your findings by each of the 3 sites

*It would also help to run a summary that shows codes/sub codes by caregiver characteristics

Then do a comparative analysis. I find it easier to use a visual- a summary matrix during my data management and analysis. For such a study, it would have 39 columns to capture all the codes/sub-codes from all the data collection sessions. It makes comparison easier.

-Edit grammar in this section.

RESULTS

-Ensure all content provided in Table 1 is reported in narrative format. This is still pending. This guideline applies where all tables are concerned.

- You continue to provide sections of transcripts as data findings. This is still pending.

Do findings differ by sites? FGDs? KIIs? Demographics - especially care giver characteristics?

DISCUSSION

-This section has improved. Thank you

-Once you revise the results section, refine the discussion and conclusion sections.

CONCLUSION

-Add a conclusion in this manuscript.

7. PLOS authors have the option to publish the peer review history of their article (what does this mean?). If published, this will include your full peer review and any attached files.

Reviewer #2: **Yes: **VIOLET NAANYU

---

## [Author Response · Author response to Decision Letter 1]

11 Dec 2023

1. In your abstract, as you report findings, highlight what was reported by both groups (caregivers, healthcare providers) and then what was unique to each group. This is still pending.

We have specified which results came from which participant group.

2. Consider having a conclusion, not a discussion section in the abstract.

We have changed discussion to “conclusion” in the abstract.

METHODS

3. Your study used other languages besides English. Add information on informed consent, translation and associated consenting documents. This is still pending.

We have specified that: “Participants provided written informed consent in compliance with IRB approvals at the University of Kansas Medical Center (STUDY00147024) and University of Nairobi (P457/06/2021). Caregiver informed consent documents and FGD guides were available in English, Luo, and Swahili and all documents and translations were approved by the IRB at the University of Nairobi. Provider informed consent documents and interview guides were available in English only, as all healthcare providers were fluent in English. During the informed consent process, the interviewer explained the purpose of the study, risks, and benefits of the study, and emphasized that study participant was optional. Immediately after informed consent was provided but prior to the focus groups or KIIs, surveys with each participant were conducted to capture socio-demographics characteristics of providers and caregivers. Informed consent interviews and surveys occurred in a private area of the hospital and were conducted by a research assistant in English, Swahili, or Luo, per the participant’s preference. “ We have added a line that specifies caregiver documents were available in 3 languages and that all documents and translations were approve by the IRB at a local institution.

4. It is still not clear which findings come from which of the 9 FGDs and/or which of the KIIs. I suggest you get 3 index refereed articles to guide you in your data analyses and reporting. You need to:

*Run analysis for the 9 FGDs

*Run analysis for each of the 30 KIIs

*Run a summary that shows your findings by each of the 3 sites

*It would also help to run a summary that shows codes/sub codes by caregiver characteristics

Then do a comparative analysis. I find it easier to use a visual- a summary matrix during my data management and analysis. For such a study, it would have 39 columns to capture all the codes/sub-codes from all the data collection sessions. It makes comparison easier.

The objective of this study was to assess general challenges to pediatric ART in Kenya to inform development of a product that could overcome these challenges. While we included multiple caregiver types to represent the range of caregivers who provide care to children living with HIV, we did not intend to analyze differences between caregiver types. That said, there was great consensus across the concerns raised. We feel that such a quantitative approach to this qualitative study is unnecessary given the consensus. 

5. Edit grammar in this section.

We have edited the grammar to standard English.

RESULTS

6. Ensure all content provided in Table 1 is reported in narrative format. This is still pending. This guideline applies where all tables are concerned.

The paragraph proceeding the tables summarizes key items in the table without being redundant of the information in the table. We feel that adding more descriptive text would not add meaningful content to the manuscript and would be duplicative of the information provided in the table.

7. You continue to provide sections of transcripts as data findings. This is still pending.

Per the COREQ guidelines, which we have attached as a supplementary material and complies with journal requirements, item 29 requires participant quotations to highlight themes and findings. We have opted to retain these quotations as required by journal guidelines and as the accepted practice in qualitative data presentation.

8. Do findings differ by sites? FGDs? KIIs? Demographics - especially care giver characteristics?

Per response above: The objective of this study was to assess general challenges to pediatric ART in Kenya to inform development of a product that could overcome these challenges. While we included multiple caregiver types to represent the range of caregivers who provide care to children living with HIV, we did not intend to analyze differences between caregiver types. That said, there was great consensus across the concerns raised. We feel that such a quantitative approach to this qualitative study is unnecessary given the consensus. 

DISCUSSION

9. This section has improved. Thank you

Once you revise the results section, refine the discussion and conclusion sections.

We have opted not to revise the results section, as discussed above.

CONCLUSION

10. Add a conclusion in this manuscript.

We have added a conclusion to this manuscript.

---

## [Editor Report · Decision Letter 2]

18 Dec 2023

Challenges with pediatric antiretroviral therapy administration: qualitative perspectives from caregivers and HIV providers in Kenya

PONE-D-23-14403R2

Dear Dr. Wexler,

We’re pleased to inform you that your manuscript has been judged scientifically suitable for publication and will be formally accepted for publication once it meets all outstanding technical requirements.

Kind regards,

Petros Isaakidis MD, PhD

Academic Editor

PLOS ONE

Additional Editor Comments (optional):

I fully support your decision to maintain the quotes in the manuscript (excellent response), and I also did appreciate your response about the comparisons between groups.
---

## [Editor Report · Acceptance letter]

28 Dec 2023

PONE-D-23-14403R2 

PLOS ONE

Dear Dr. Wexler, 

I'm pleased to inform you that your manuscript has been deemed suitable for publication in PLOS ONE. Congratulations! Your manuscript is now being handed over to our production team.

Kind regards, 

on behalf of

Dr. Petros Isaakidis 

Academic Editor

PLOS ONE